# Potential of Hematologic Parameters in Predicting Mortality of Patients with Traumatic Brain Injury

**DOI:** 10.3390/jcm11113220

**Published:** 2022-06-05

**Authors:** Sol Bi Kim, Youngjoon Park, Ju Won Ahn, Jeongmin Sim, Jeongman Park, Yu Jin Kim, So Jung Hwang, Kyoung Su Sung, Jaejoon Lim

**Affiliations:** 1Department of Neurosurgery, Bundang CHA Medical Center, CHA University, Yatap-dong 59, Seongnam 13496, Korea; a186051@chamc.co.kr (S.B.K.); eugene@chauniv.ac.kr (J.W.A.); simti123@chauniv.ac.kr (J.S.); jungman.park@chauniv.ac.kr (J.P.); petaldew17@chauniv.ac.kr (Y.J.K.); sjhwang7@chamc.co.kr (S.J.H.); 2Department of Biomedical Science, College of Life Science, CHA University, Seongnam 13488, Korea; yjparkep@chauniv.ac.kr; 3Department of Neurosurgery, Dong-A University Hospital, Dong-A University College of Medicine, Busan 49201, Korea

**Keywords:** brain injury, mortality, prediction model, trauma

## Abstract

Traumatic brain injury (TBI) occurs frequently, and acute TBI requiring surgical treatment is closely related to patient survival. Models for predicting the prognosis of patients with TBI do not consider various factors of patient status; therefore, it is difficult to predict the prognosis more accurately. In this study, we created a model that can predict the survival of patients with TBI by adding hematologic parameters along with existing non-hematologic parameters. The best-fitting model was created using the Akaike information criterion (AIC), and hematologic factors including preoperative hematocrit, preoperative C-reactive protein (CRP), postoperative white blood cell (WBC) count, and postoperative hemoglobin were selected to predict the prognosis. Among several prediction models, the model that included age, Glasgow Coma Scale, Injury Severity Score, preoperative hematocrit, preoperative CRP, postoperative WBC count, postoperative hemoglobin, and postoperative CRP showed the highest area under the curve and the lowest corrected AIC for a finite sample size. Our study showed a new prediction model for mortality in patients with TBI using non-hematologic and hematologic parameters. This prediction model could be useful for the management of patients with TBI.

## 1. Introduction

Traumatic brain injury (TBI) occurs frequently and has a significant impact on patient functional outcomes. TBI can be mild, moderate, or severe based on the patient’s status [1]. Neurosurgical treatment should be considered in moderate and severe TBI. Moderate and severe TBI are also closely related to poor survival outcomes and high mortality; therefore, predicting survival could be important for patient treatment and prognosis [2,3,4]. In the 1980s, the Trauma and Injury Severity Score (TRISS), which was calculated using the Injury Severity Score (ISS), was developed and used as a gold standard for predicting mortality in patients with TBI [5,6,7]. However, the ISS has poor accuracy in predicting mortality in patients with moderate and severe TBI [8,9]. In many subsequent studies, it has been reported that hematologic status, which has not been evaluated in ISS, has an important association with prognosis, especially survival outcomes [10,11]. We assessed whether hematologic and non-hematologic parameters could be factors in predicting the mortality of patients with TBI. This study aimed to create a model to predict the survival of surgically treated patients with moderate and severe TBI, including hematologic and non-hematologic parameters.

## 2. Materials and Methods

### 2.1. Inclusion and Exclusion Criteria of Participants

From January 2005 to December 2019, data from 1539 patients with TBI treated with surgery were collected from the Bundang CHA Medical Center. Only patients with acute TBI were included in this study. Patients treated within one week of TBI were classified as acute, and those treated after one week were classified as chronic. Patients with chronic TBI (n = 821) were excluded. Because the surgically treated TBI patient cohort groups were heterogeneous, we only included open craniotomy treated TBI patients. Patients with burr-hole trephination (n = 112) or stereotaxic catheter insertion (n = 63) were also excluded. In addition, we excluded patients who did not have information on hematologic and non-hematologic parameters (n = 54). Finally, surgically treated 489 patients with moderate and severe TBI were included in the study (Figure 1). This study was approved by the Institutional Review Board of the Bundang CHA Medical Center.

### 2.2. Clinical Information and Relevance

Pre- and postoperative computed tomography (CT) scans were reviewed by two neuroradiologists. Additional variables obtained for analysis included age, height, weight, sex, Glasgow Coma Scale (GCS) score, ISS, overall survival, and hematologic parameters. Preoperative and postoperative common blood test values (WBC, hemoglobin, hematocrit, Platelets, RDW, MPC, MCV, MCH, MCHC, CRP, Creatine) were obtained as a hematologic parameter. Survival outcomes were analyzed by considering these factors.

### 2.3. Statistical Analysis and Model Development

The *t*-test and chi-squared test were performed to determine the clinical and hematological parameters that differed in survival over 30 days. Multiple logistic regression analysis was performed with all parameter combinations to estimate the optimal slope of the clinical and hematological parameters. We selected the best-fitting model with a minimum Akaike information criterion (AIC) and corrected AIC for finite sample size (AICc) value using the R package ‘leaps’ (R Foundation, Vienna, Austria). The best prediction model with five hematologic parameters was established using the following formula:

If the *i*th clinical parameter and estimate standard by multiple logistic regression analysis are $X_{i}$ and $\beta_{i}$, respectively, then the blood prediction model (BPM) equation can be expressed as follows:(1)$$\begin{array}{c} Ps=\beta_{0}+\beta_{1}X_{1}+\beta_{2}X_{2}\ldots \;\beta_{i}X_{i},\\ BPM=\frac{1}{1+e^{-Ps}}\;\ldots \; \end{array}$$

### 2.4. Model Validation

To evaluate the performance of the best prediction model with five hematologic parameters, we compared the discrimination and calibration of all models that were combinations of non-hematologic parameters. We assessed the Hosmer–Lemeshow test statistic and area under the curve (AUC) of the receiver operating characteristic curve (ROC) for calibration and discrimination, respectively. Bias-corrected 95% confidence intervals were calculated for the AUC by resampling the bootstrapping algorithm 1000 times.

## 3. Results

Recent trauma studies have used 30-day mortality as a reasonable endpoint [12,13,14]. Death more than 30 days after trauma is considered more related to comorbidities [12]. Therefore, our study used 30-day mortality as the endpoint. We analyzed both pre- and postoperative parameters to determine the best hematologic parameters for surgically treated patients.

### 3.1. Non-Hematologic Parameters on 30-Day Mortality

Age, height, weight, sex, GCS, and ISS were used as non-hematologic parameters. Among these parameters, age, GCS, and ISS were significantly different between the mortality periods (Table 1). The long survival group (LSG) was significantly older (mean 54.38) than the short survival group (SSG) (mean 46.69) (*p* < 0.001). The GCS score was significantly higher in the LSG (mean 9.72) than in the SSG (mean 6.28) (*p* < 0.001). The ISS was significantly lower in the LSG (mean 17.69) than in the SSG (mean 64) (*p* < 0.001). In contrast, height, weight, and sex were not associated with 30-day mortality.

### 3.2. Hematologic Parameters on 30-Day Mortality

We analyzed whether the pre- or postoperative blood test parameters differed according to the survival of patients with TBI. A total of 11 common blood test values in each pre- or postoperative period were analyzed according to 30-day survival (Table 2). Pre- and postoperative red cell distribution width (RDW) was significantly lower in the LSG than in the SSG (*p* = 0.0157 and 0.0147, respectively). The postoperative mean platelet volume (MPV) was significantly higher in the LSG than in the SSG (*p* = 0.008). Pre- and postoperative hemoglobin were significantly higher in the LGS than in the SSG (both *p* < 0.001). Pre- and postoperative hematocrit levels were significantly higher in the LSG than in the SSG (*p* = 0.0012 and <0.001, respectively). Pre- and postoperative platelets were significantly higher in the LSG than in the SSG (both *p* < 0.01). Pre- and postoperative C-reactive protein (CRP) levels were significantly lower in the LSG than in the SSG (*p* < 0.001 and 0.055, respectively). The postoperative creatinine level was significantly lower in the LSG than in the SSG (*p* = 0.023). Pre- and postoperative mean corpuscular volumes (MCV) were significantly lower in the LSG than in the SSG (*p* < 0.001 and 0.003, respectively). Preoperative mean corpuscular hemoglobin (MCH) was significantly higher in the LSG than in the SSG (*p* < 0.001). The preoperative mean corpuscular hemoglobin concentration (MCHC) was significantly higher in the LSG than in the SSG (*p* = 0.021).

### 3.3. Prediction Model with Pre- and Postoperative Hematologic and Non-Hematologic Parameters

To obtain the best-fitting model, we calculated the AIC with all models that were established by multiple logistic regression and selected the model with the minimum AIC. The model with the minimum AIC contained non-hematologic parameters, including age, GCS, and ISS, and five hematologic parameters, including preoperative hematocrit, preoperative CRP, postoperative WBC count, postoperative hemoglobin, and postoperative CRP (Table 3). The coefficients of age, GCS, ISS, preoperative hematocrit, postoperative WBC count, preoperative CRP, preoperative hemoglobin, and preoperative CRP were 0.048, −0.434, 0.103, 0.398, −0.115, −0.111, −0.815, and 0.171, respectively (Table 3).

### 3.4. Performance of the Selected Prediction Model

To evaluate the discrimination performance of the selected prediction model with hematologic parameters, we compared the AUCs of the ROC curves between the selected prediction model with hematologic parameters and the seven non-hematologic parameters (Table 4, Figure 2). The selected prediction model (age + GCS + ISS + preoperative hematocrit + preoperative CRP + postoperative WBC count + postoperative hemoglobin + postoperative CRP) had the highest AUC value (92.53) and the lowest AICc (110.868) compared with other non-hematologic models. The age + GCS prediction model had the second highest AUC (84.2), and the GCS prediction model had the third highest AUC (83.85) (Table 4).

## 4. Discussion

Our study showed that the performance of the selected prediction model with hematologic parameters was better than that of other non-hematologic models. Five hematologic parameters (preoperative hematocrit, preoperative CRP, postoperative WBC, postoperative hemoglobin, and postoperative CRP) were used to obtain the best-fitting model. Additionally, among the non-hematologic parameters, age, GCS, and ISS levels were significantly different between the two mortality periods.

Several studies have shown that hematologic factors are associated with the prognosis of TBI [15,16,17,18,19,20,21]. It is important to avoid hypoxia to prevent secondary brain injury in patients with TBI [22]. For theoretical increases in oxygen-carrying capacity, maintaining a hematocrit above 30% is recommended for patients with TBI [23]. Several studies have shown an association between hemoglobin, hematocrit, and prognosis in patients with TBI. Salim et al. reported that anemia was a significant risk factor for mortality (adjusted odds ratio (AOR), 1.59; 95% confidence interval (CI), 1.13 to 2.24; *p* = 0.007) and complications (AOR, 1.95; 95% CI, 1.42 to 2.70; *p* < 0.001) in patients with TBI [19]. Zhou et al. reported that after being adjusted to predict patient survival, the combination of postoperative hematocrit and change in hematocrit demonstrated the highest sensitivity (77.5%) and specificity (89.4%), and the best accuracy was 94.5% when used to predict prognosis for these patients [21]. The selected prediction model (age + GCS + ISS + preoperative hematocrit + preoperative CRP + postoperative WBC count + postoperative hemoglobin + postoperative CRP) was developed by considering not only previously identified important factors for predicting TBI outcome, but also hematologic factors that can accurately reflect the pre- and postoperative status of patients with moderate to severe TBI who underwent neurosurgery treatment. As a result, it is thought to be more accurate than the prior model at predicting the patient’s prognosis, particularly the 30-day mortality, which is a crucial period for the acute TBI.

Inflammation can result in secondary brain injury, tissue damage, and neurodegeneration [24]. Under normal conditions, the blood–brain barrier (BBB) separates the central nervous system from the blood stream. After TBI, the BBB quickly breaks. Serum components and blood cells leak into the cerebral tissue, initiating a cascade of molecular events leading to immunoactivity. The neurotoxicity of some inflammatory mediators induces neuronal cell death [25]. Rovlias et al. reported that patients with severe head injury had significantly higher WBC counts than those with moderate or minor injury (*p* < 0.001), and WBC counts were significantly higher in those with an unfavorable outcome (*p* < 0.001) [20]. In our study, postoperative WBC count and CRP level were selected to obtain the best-fitting model.

TRISS is based on patient age, ISS, and Trauma Score (TS), and is widely used in the trauma community [3]. Several studies have shown that TRISS distinguishes between survivors and non-survivors; however, it is insufficient for predictive reliability [26,27,28,29]. TRISS is a poor predictor of multiple severe traumas in one region [30]. The GCS score, which is incorporated into TRISS, can change during the early phase of trauma with changes in consciousness [31,32,33]. There are inaccuracies in GCS score calculations even among doctors [31,34]. However, using general hematological parameters, our model can be more objective.

Our study had several limitations. There could be confounding factors because this was a retrospective study and the subject size was not large. Surgeons’ skills may influence the outcome. However, our study could be significant in terms of using general hematological parameters for predicting mortality, and these factors could assist physicians in managing patients and making decisions.

## 5. Conclusions

Our study showed a new prediction model for mortality in patients with TBI using non-hematologic and hematologic parameters. This prediction model could be useful in the management of patients with TBI.

## Figures and Tables

**Figure 1 jcm-11-03220-f001:**
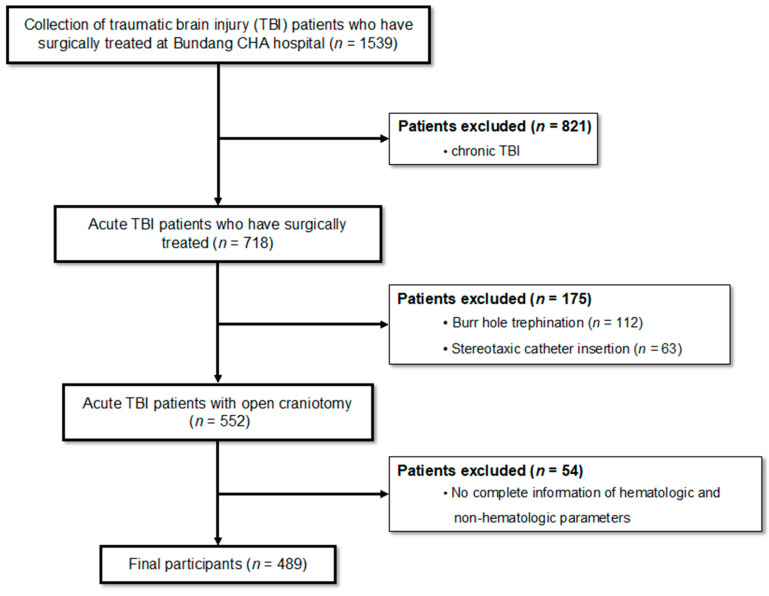
Inclusion and exclusion criteria of participants. Data from 1539 patients with TBI treated by surgery were collected. Patients with chronic TBI (n = 821) were excluded. Only patients with acute TBI were included in this study. Because the surgically treated TBI patient cohort groups were heterogeneous, we only included open craniotomy treated TBI patients. Patients with burr-hole trephination (n = 112) or stereotaxic catheter insertion (n = 63) were also excluded. In addition, we excluded patients who did not have information on hematologic and non-hematologic parameters (n = 54). Finally, surgically treated 489 patients with moderate and severe TBI were included in the study. TBI, traumatic brain injury.

**Figure 2 jcm-11-03220-f002:**
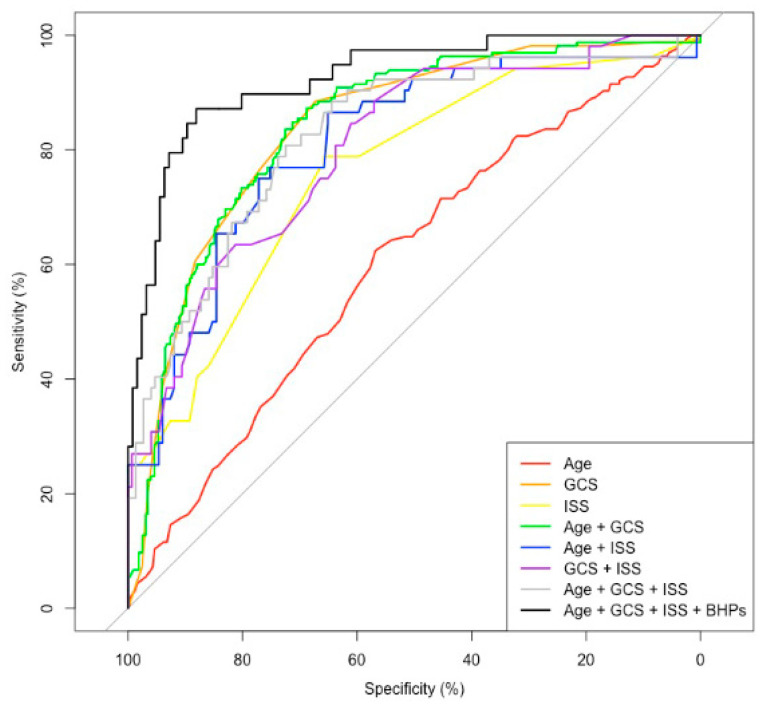
Performance of the selected prediction model. Performance of the selected prediction model with hematologic parameters. We compared the AUC of the ROC curve between the selected prediction model with hematologic parameters and the seven non-hematologic parameters. The selected prediction model (age + GCS + ISS + preoperative hematocrit + preoperative CRP + postoperative WBC count + postoperative hemoglobin + postoperative CRP) had the highest AUC compared to other non-hematologic models. The age + GCS prediction model had the second highest AUC, and the GCS prediction model had the third highest AUC. AUC, area under the curve; ROC, receiver operating characteristic; GCS, Glasgow Coma Scale; ISS, Injury Severity Scale; CRP, C-reactive protein; WBC, white blood cell.

**Table 1 jcm-11-03220-t001:** Statistical analysis with non-hematologic parameters on 30-day mortality.

	Long Survival Group	Short Survival Group	*p*-Value
Age, n (mean)	324 (46.69 years)	165 (54.38 years)	<0.001
Height, n (mean)	324 (166.92 cm)	165 (163.08 cm)	0.4488
Weight, n (mean)	324 (60.29 kg)	165 (61.04 kg)	0.6028
Sex (n)			
Male	248	119	
Female	76	46	
			0.3381
ISS, n (mean)	149 (17.69)	52 (34)	<0.001
GCS, n (mean)	324 (9.72)	165 (6.28)	<0.001

Long survival group: survival longer than 30 days. Short survival group: survival shorter than 30 days. n, number of patients; ISS, Injury Severity Score; GCS, Glasgow Coma Scale.

**Table 2 jcm-11-03220-t002:** Statistical analysis hematologic parameters on 30-day mortality.

	Long Survival Group	Short Survival Group	*p*-Value	*p* Adj
Preoperative, n (Mean)				
RDW	321 (13.58%)	164 (14.02%)	0.016	0.346
MPV	312 (8.75 fL)	162 (8.48 fL)	0.037	0.822
WBC	321 (13.36 × 10^3^/uL)	164 (13.93 × 10^3^/uL)	0.367	1.000
Hemoglobin	322 (12.98 g/dL)	164(12.19 g/dL)	<0.001	0.012
Hematocrit	322 (37.87%)	164 (35.8%)	0.002	0.035
Platelets	321 (222.44 × 10^3^/uL)	164 (192.74 × 10^3^/uL)	<0.001	0.017
CRP	288 (7.88 mg/dL)	122 (12.25 mg/dL)	<0.001	0.009
Creatinine	322 (0.94 mg/dL)	163 (1.13 mg/dL)	0.046	1
MCV	321 (91 fL)	164 (93.7 fL)	<0.001	<0.001
MCH	321 (31.16 pg)	164 (31.92 pg)	<0.001	0.015
MCHC	321 (34.24 g/dL)	164 (34.06 g/dL)	0.021	0.467
Postoperative, n (Mean)				
RDW	321 (13.87%))	162 (14.27%))	0.015	0.323
MPV	312 (8.7 fL)	160 (8.35 fL)	0.008	0.166
WBC	321 (14.02 × 10^3^/uL)	162 (14.19 × 10^3^/uL)	0.632	1.000
Hemoglobin	324 (11.98 g/dL)	162 (11.17 g/dL)	<0.001	0.004
Hematocrit	324 (34.92%)	162 (32.87%)	<0.001	0.021
Platelets	324 (183.84 × 10^3^/uL)	162 (139.11 × 10^3^/uL)	<0.001	<0.001
CRP	153 (8.09 mg/dL)	62 (10.84 mg/dL)	0.055	1.000
Creatinine	324 (0.86 mg/dL)	160 (1.15 mg/dL)	0.023	0.503
MCV	321 (90.69 fL)	162 (92.14 fL)	0.003	0.055
MCH	321 (31.1 pg)	162 (31.44 pg)	0.060	1.000
MCHC	321 (34.29 g/dL)	162 (34.13 g/dL)	0.039	0.867

Long survival group: survival longer than 30 days. Short survival group: survival shorter than 30 days. n, number of patients. RDW; red blood cell width distribution, MPV; mean platelet volume, WBC; white blood cell count, CRP; C-reactive protein, MCV; mean corpuscular volume, MCH; mean corpuscular hemoglobin, MCHC; mean corpuscular hemoglobin concentration, Bold; significant results, *p* adj; Adjusted *p*-value.

**Table 3 jcm-11-03220-t003:** Best prediction model parameters by multiple logistic regression.

Parameter	Coefficient	Std. Error	Z-Statics	*p*-Value
Intercept	−7.621	3.293	−2.314	0.021
Age	0.048	0.020	2.391	0.017
GCS	−0.434	0.128	−3.401	0.001
ISS	0.103	0.033	3.133	0.002
Pre-Hct	0.398	0.115	3.450	0.001
Post-WBC	−0.115	0.061	−1.904	0.057
Pre-CRP	−0.111	0.069	−1.605	0.108
Post-Hgb	−0.815	0.272	−2.996	0.003
Post-CRP	0.171	0.071	2.410	0.016

GCS, Glasgow Coma Scale; Hct, hematocrit; ISS, Injury Severity Score; Std. error, standard error; Post, postoperative hematologic value; Pre, pre-operative hematologic value; WBC, white blood cell; CRP, C-reactive protein; Hgb, hemoglobin.

**Table 4 jcm-11-03220-t004:** Selected prediction model performance for 30 days mortality with best prediction parameters.

Prediction Model	AUC (CI 95%)	Adj. AUC	AIC	AICc	HL (Statistic)	HL (*p*-Value)
Age	60.32 (55.06–65.59)	60.205	615.349	615.358	8.479	0.388
GCS	83.85 (80.16–87.54)	83.815	465.127	465.135	-	-
ISS	76.06 (68.53–83.6)	76.015	188.433	188.453	3.845	0.871
Age + GCS	84.2 (80.55–87.85)	84.115	463.669	463.694	9.149	0.330
Age + ISS	80.96 (73.91–88.02)	80.435	182.128	182.189	11.196	0.191
GCS + ISS	80.19 (73.32–87.07)	79.900	182.356	182.417	11.622	0.169
Age + GCS + ISS	82.6 (75.83–89.38)	81.825	177.760	177.882	8.937	0.348
Age + GCS + ISS + BHPs	92.53 (87.84–97.22)	90.045	109.944	110.868	8.468	0.389

AUC, area under the curve; CI, confidence interval; Adj. AUC, bias-corrected c-index (AUC) by re-sampling with bootstrap method (n = 1000); AIC, Akaike information criterion; AICc, corrected AIC for finite sample sizes; HL, Hosmer–Lemeshow test; GCS, Glasgow Coma Scale; ISS, Injury Severity Score; BHPs, best hematologic prediction parameters.

## Data Availability

Not applicable.

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
