# Peer review of "Potential of Hematologic Parameters in Predicting Mortality of Patients with Traumatic Brain Injury"

_jcm, 2022, doi:10.3390/jcm11113220_

Round 1
Reviewer 1 Report
The use of hematologic parameters in predicting mortality of patients with traumatic brain injury is an interesting idea, which deserves attention. Authors included these parameters in prediction models and obtained promising results. There are some minor suggestions that could be considered to improve the manuscript:
- Some numbers are presented without measurement units (eg Table 1 and 2).
- Titles of the tables should be more descriptive. Some legends are incomplete.
- The number of decimal places for data presented in tables (eg Table 1) is surprisingly large. How can authors justify such precision? Can they give any confidence interval to show this high precision?
4. Hematologic parameters are included in only one prediction model. Is there any specific reason for not including them in other models (eg Age + GCS).
Author Response
Reviewer #1
The use of hematologic parameters in predicting mortality of patients with traumatic brain injury is an interesting idea, which deserves attention. Authors included these parameters in prediction models and obtained promising results. There are some minor suggestions that could be considered to improve the manuscript:
- Some numbers are presented without measurement units (eg Table 1 and 2).
Response: We appreciate the reviewer’s thoughtful suggestion. The draft was revised according to the reviewer's comments.
- Titles of the tables should be more descriptive. Some legends are incomplete.
Response: We appreciate the reviewer’s point. The draft was revised in table and figure section according to the reviewer's comments.
- The number of decimal places for data presented in tables (eg Table 1) is surprisingly large. How can authors justify such precision? Can they give any confidence interval to show this high precision?
Response: We appreciate the reviewer’s point. The number of digits increased while calculating the average of each value. We corrected for rounding to 2 decimal places.
- Hematologic parameters are included in only one prediction model. Is there any specific reason for not including them in other models (eg Age + GCS).
Response: We appreciate the reviewer’s s critical points. Along with AUC, we also considered the Akaike information criterion (AIC) value, which is an important measure of prediction error. Although the model of Age + ISS + GCS is lower than that of Age + GCS or GCS in terms of AUC, hematologic parameters were applied to the Age + GCS + ISS model with a low AIC value.

Reviewer 2 Report
Paper is well written and well organized. Some points:
- Lines 111-112: "A total of 11 common blood test values..." Please state which are these "11 common blood test values" in methods section.
- Line 45: "This study aimed to investigate a model to predict.." To investigate or to create a model ?
- Lines 35-36: "Moderate and severe TBI are also closely related to poor survival outcomes..." But also high mortality. Refs: doi: 10.3389/fneur.2021.727754 -- doi: 10.25259/SNI_697_2020
- Lines 113-114: "As a result, all parameters except the white blood cell (WBC) count were significantly different at least once in groups divided by mortality period and pre- or post-operation" This is quite normal. What did author want to say?
- Lines 104-105: "The GCS score was significantly higher in the LSG (mean 9.72) than in the SSG (mean 6.28)" Can this represent a bias for this study?
- Lines 168-169: "Several studies have shown that hematologic factors are associated with the prognosis of TBI [13-19]" What does this paper add new to the literature?
- About mild, moderate and severe TBI. Which are the inclusion criteria ?
- Please report why "Age + GCS + ISS + BHPs" is more accurate than other in your opinion.
Author Response
Reviewer #2
Paper is well written and well organized. Some points:
- Lines 111-112: "A total of 11 common blood test values..." Please state which are these "11 common blood test values" in methods section.
Response: We appreciate the Reviewer’s thoughtful comments. The draft was revised according to the reviewer's comments.
- Line 45: "This study aimed to investigate a model to predict" To investigate or to create a model?
Response: We appreciate the Reviewer’s thoughtful advice. This study aimed to create a model to predict the survival of surgically treated with moderate and severe TBI. The draft was revised according to the reviewer's comments.
- Lines 35-36: "Moderate and severe TBI are also closely related to poor survival outcomes..." But also high mortality. Refs: doi: 10.3389/fneur.2021.727754 -- doi: 10.25259/SNI_697_2020
Response: We appreciate the Reviewer’s points. The draft was revised according to the reviewer's comments and references were added.
- Lines 113-114: "As a result, all parameters except the white blood cell (WBC) count were significantly different at least once in groups divided by mortality period and pre- or post-operation" This is quite normal. What did author want to say?
Response: We agree with the Reviewer’s points. According to the reviewer's opinion, the sentence "As a result, all parameters except the white blood cell (WBC) count were significantly different at least once in groups divided by mortality period and pre- or post-operation" was deleted from the draft.
- Lines 104-105: "The GCS score was significantly higher in the LSG (mean 9.72) than in the SSG (mean 6.28)" Can this represent a bias for this study?
Response: We appreciate the reviewer’s point. Low Glasgow Coma Scale (GCS) scores in patients with traumatic brain injury (TBI) indicate a poor prognosis in several studies1,2,3,4. In this study, similar to previous studies, the GCS score was significantly higher in the long survival group (LSG) than the short survival group (SSG). To obtain the best-fitting model, we calculated the AIC with all models that were established by multiple logistic regression and selected the model with the minimum AIC. The model with the minimum AIC contained GCS parameter.
References
- Signorini DF, Andrews PJ, Jones PA,Wardlaw JM, Miller JD. Predicting survival using simple clinical variables: a case study in traumatic brain injury. J Neurol Neurosurg Psychiatry 1999; 66:20–5.
- Lieberman JD, Pasquale MD, Garcia R, Cipolle MD, Mark Li P, Wasser TE. Use of admission Glasgow Coma Score, pupil size, and pupil reactivity to determine outcome for trauma patients. J Trauma 2003;55:437–42.
- Pal J, Brown R, Fleiszer D. The value of the Glasgow Coma Scale and Injury Severity Score: predicting outcome in multiple trauma patients with head injury. J Trauma. 1989;29:746–748.
- Farid S, Amar J, Melinda M, Ammar S, Jacklyn O, Steven T. Is it possible to recover from traumatic brain injury and a Glasgow coma scale score of 3 at emergency department presentation? Am J Emerg Med . 2018 Sep;36(9):1624-1626.
- Lines 168-169: "Several studies have shown that hematologic factors are associated with the prognosis of TBI [13-19]" What does this paper add new to the literature?
Response: We appreciate the Reviewer’s thoughtful comment. Several studies have shown that each hematologic factors (WBC, Hematocrit, CRP, Hemoglobin) are associated with the prognosis of TBI. Our prediction model contained non-hematologic parameters (age, GCS, ISS) and five hematologic parameters (preoperative hematocrit, preoperative CRP, postoperative WBC, postoperative hemoglobin, postoperative CRP).
- About mild, moderate, and severe TBI. Which are the inclusion criteria?
Response: We appreciate the Reviewer’s points. This study aimed to create a model to predict the survival of surgically treated patients with moderate and severe TBI. As the reviewer correctly points out, we revised the manuscript. Finally, surgically treated 489 patients with moderate and severe TBI were included in this study.
- Please report why "Age + GCS + ISS + BHPs" is more accurate than other in your opinion.
Response: We appreciate the Reviewer’s thoughtful comment. This "Age + GCS + ISS + BHPs" prediction model was developed by considering not only previously identified important factors for predicting TBI outcome, but also hematologic factors that can accurately reflect the pre- and post-operative status of patients with moderate to severe TBI who underwent neurosurgery treatment. As a result, it is thought to be more accurate than the prior model at predicting the patient's prognosis, particularly the 30-day motility, which is a crucial period for the acute TBI. The draft was revised in discussion section according to the reviewer's comments

Round 2
Reviewer 2 Report
Authors solved all my criticisms.